# LEARNING TO DISCRETIZE: SOLVING 1D SCALAR CONSERVATION LAWS VIA DEEP REINFORCEMENT LEARNING

## ABSTRACT

Conservation laws are considered to be fundamental laws of nature. It has broad application in many fields including physics, chemistry, biology, geology, and engineering. Solving the differential equations associated with conservation laws is a major branch in computational mathematics. Recent success of machine learning, especially deep learning, in areas such as computer vision and natural language processing, has attracted a lot of attention from the community of computational mathematics and inspired many intriguing works in combining machine learning with traditional methods. In this paper, we are the first to view numerical PDE solvers as a MDP and to use (deep) RL to learn new solvers. As a proof of concept, we focus on 1-dimensional scalar conservation laws. We deploy the machinery of deep reinforcement learning to train a policy network that can decide on how the numerical solutions should be approximated in a sequential and spatial-temporal adaptive manner. We will show that the problem of solving conservation laws can be naturally viewed as a sequential decision making process and the numerical schemes learned in such a way can easily enforce long-term accuracy. Furthermore, the learned policy network is carefully designed to determine a good local discrete approximation based on the current state of the solution, which essentially makes the proposed method a meta-learning approach. In other words, the proposed method is capable of learning how to discretize for a given situation mimicking human experts. Finally, we will provide details on how the policy network is trained, how well it performs compared with some state-of-the-art numerical solvers such as WENO schemes, and how well it generalizes. Our code is released anomynously at `https://github.com/qwerlanksdf/L2D`.

## 1 INTRODUCTION

Conservation laws are considered to be one of the fundamental laws of nature, and has broad applications in multiple fields such as physics, chemistry, biology, geology, and engineering. For example, Burger's equation, a very classic partial differential equation (PDE) in conservation laws, has important applications in fluid mechanics, nonlinear acoustics, gas dynamics, and traffic flow.

Solving the differential equations associated with conservation laws has been a major branch of computational mathematics (LeVeque, 1992; 2002), and a lot of effective methods have been proposed, from classic methods such as the upwind scheme, the Lax-Friedrichs scheme, to the advanced ones such as the ENO/WENO schemes (Liu et al., 1994; Shu, 1998), the flux-limiter methods (Jerez Galiano & Uh Zapata, 2010), and etc. In the past few decades, these traditional methods have been proven successful in solving conservation laws. Nonetheless, the design of some of the high-end methods heavily relies on expert knowledge and the coding of these methods can be a laborious process. To ease the usage and potentially improve these traditional algorithms, machine learning, especially deep learning, has been recently incorporated into this field. For example, the ENO scheme requires lots of 'if/else' logical judgments when used to solve complicated system of equations or high-dimensional equations. This very much resembles the old-fashioned expert systems. The recent trend in artificial intelligence (AI) is to replace the expert systems by the so-called 'connectionism', e.g., deep neural networks, which leads to the recent bloom of AI. Therefore, it

is natural and potentially beneficial to introduce deep learning in traditional numerical solvers of conservation laws.

## 1.1 RELATED WORKS

In the last few years, neural networks (NNs) have been applied to solving ODEs/PDEs or the associated inverse problems. These works can be roughly classified into three categories according to the way that the NN is used.

The first type of works propose to harness the representation power of NNs, and are irrelevant to the numerical discretization based methods. For example, Raissi et al. (2017a;b); Yohai Bar-Sinai (2018) treated the NNs as new ansatz to approximate solutions of PDEs. It was later generalized by Wei et al. (2019) to allow randomness in the solution which is trained using policy gradient. More recent works along this line include (Magiera et al., 2019; Michoski et al., 2019; Both et al., 2019). Besides, several works have focused on using NNs to establish direct mappings between the parameters of the PDEs (e.g. the coefficient field or the ground state energy) and their associated solutions (Khoo et al., 2017; Khoo & Ying, 2018; Li et al., 2019; Fan et al., 2018b). Furthermore, Han et al. (2018); Beck et al. (2017) proposed a method to solve very high-dimensional PDEs by converting the PDE to a stochastic control problem and use NNs to approximate the gradient of the solution.

The second type of works focus on the connection between deep neural networks (DNNs) and dynamic systems (Weinan, 2017; Chang et al., 2017; Lu et al., 2018; Long et al., 2018b; Chen et al., 2018). These works observed that there are connections between DNNs and dynamic systems (e.g. differential equations or unrolled optimization algorithms) so that we can combine deep learning with traditional tools from applied and computational mathematics to handle challenging tasks in inverse problems (Long et al., 2018b;a; Qin et al., 2018). The main focus of these works, however, is to solve inverse problems, instead of learning numerical discretizations of differential equations. Nonetheless, these methods are closely related to numerical differential equations since learning a proper discretization is often an important auxiliary task for these methods to accurately recover the form of the differential equations.

The third type of works, which target at using NNs to learn new numerical schemes, are closely related to our work. However, we note that these works mainly fall in the setting of supervised learning (SL). For example, Discacciati et al. (2019) proposed to integrate NNs into high-order numerical solvers to predict artificial viscosity; Ray & Hesthaven (2018) trained a multilayer perceptron to replace traditional indicators for identifying troubled-cells in high-resolution schemes for conservation laws. These works greatly advanced the development in machine learning based design of numerical schemes for conservation laws. Note that in Discacciati et al. (2019), the authors only utilized the one-step error to train the artificial viscosity networks without taking into account the long-term accuracy of the learned numerical scheme. Ray & Hesthaven (2018) first constructed several functions with known regularities and then used them to train a neural network to predict the location of discontinuity, which was later used to choose a proper slope limiter. Therefore, the training of the NNs is separated from the numerical scheme. Then, a natural question is whether we can learn discretization of differential equations in an end-to-end fashion and the learned discrete scheme also takes long-term accuracy into account. This motivates us to employ reinforcement learning to learn good solvers for conservation laws.

## 1.2 OUR APPROACH

The main objective of this paper is to design new numerical schemes in an autonomous way. We propose to use reinforcement learning (RL) to aid the process of solving the conservation laws. **To our best knowledge, we are the first to regard numerical PDE solvers as a MDP and to use (deep) RL to learn new solvers**. We carefully design the proposed RL-based method so that the learned policy can generate high accuracy numerical schemes and can well generalize in varied situations. Details will be given in section 3.

Here, we first provide a brief discussion on the benefits of using RL to solve conservation laws (the arguments apply to general evolution PDEs as well):

- Most of the numerical solvers of conservation law can be interpreted naturally as a sequential decision making process (e.g., the approximated grid values at the current time instance definitely

affects all the future approximations). Thus, it can be easily formulated as a Markov Decision Process (MDP) and solved by RL.

- In almost all the RL algorithms, the policy $\pi$ (which is the AI agent who decides on how the solution should be approximated locally) is optimized with regards to the values $Q^\pi(s_0, a_0) = r(s_0, a_0) + \sum_{t=1}^\infty \gamma^t r(s_t, a_t)$, which by definition considers the long-term accumulated reward (or, error of the learned numerical scheme), thus could naturally guarantee the long-term accuracy of the learned schemes, instead of greedily deciding the local approximation which is the case for most numerical PDEs solvers. Furthermore, it can gracefully handle the cases when the action space is discrete, which is in fact one of the major strength of RL.

- By optimizing towards long-term accuracy and effective exploration, we believe that RL has a good potential in improving traditional numerical schemes, especially in parts where no clear design principles exist. For example, although the WENO-5 scheme achieves optimal order of accuracy at smooth regions of the solution (Shu, 1998), the best way of choosing templates near singularities remains unknown. Our belief that RL could shed lights on such parts is later verified in the experiments: the trained RL policy demonstrated new behaviours and is able to select better templates than WENO and hence approximate the solution better than WENO near singularities.

- Non-smooth norms such as the infinity norm of the error is often used to evaluate the performance of the learned numerical schemes. As the norm of the error serves as the loss function for the learning algorithms, computing the gradient of the infinity norm can be problematic for supervised learning, while RL does not have such problem since it does not explicitly take gradients of the loss function (i.e. the reward function for RL).

- Learning the policy $\pi$ within the RL framework makes the algorithm meta-learning-like (Schmidhuber, 1987; Bengio et al., 1992; Andrychowicz et al., 2016; Li & Malik, 2016; Finn et al., 2017). The learned policy $\pi$ can decide on which local numerical approximation to use by judging from the current state of the solution (e.g. local smoothness, oscillatory patterns, dissipation, etc). This is vastly different from regular (non-meta-) learning where the algorithms directly make inference on the numerical schemes without the aid of an additional network such as $\pi$. As subtle the difference as it may seem, meta-learning-like methods have been proven effective in various applications such as in image restoration (Jin et al., 2017; Fan et al., 2018a; Zhang et al., 2019). See (Vanschoren, 2018) for a comprehensive survey on meta-learning.

- Another purpose of this paper is to raise an awareness of the connection between MDP and numerical PDE solvers, and the general idea of how to use RL to improve PDE solvers or even finding brand new ones. Furthermore, in computational mathematics, a lot of numerical algorithms are sequential, and the computation at each step is expert-designed and usually greedy, e.g., the conjugate gradient method, the fast sweeping method (Zhao, 2005), matching pursuit (Mallat & Zhang, 1993), etc. We hope our work could motivate more researches in combining RL and computational mathematics, and stimulate more exploration on using RL as a tool to tackle the bottleneck problems in computational mathematics.

Our paper is organized as follows. In section 2 we briefly review 1-dimensional conservation laws and the WENO schemes. In section 3, we discuss how to formulate the process of numerically solving conservation laws into a Markov Decision Process. Then, we present details on how to train a policy network to mimic human expert in choosing discrete schemes in a spatial-temporary adaptive manner by learning upon WENO. In section 4, we conduct numerical experiments on 1-D conservation laws to demonstrate the performance of our trained policy network. Our experimental results show that the trained policy network indeed learned to adaptively choose good discrete schemes that offer better results than the state-of-the-art WENO scheme which is 5th order accurate in space and 4th order accurate in time. This serves as an evidence that the proposed RL framework has the potential to design high-performance numerical schemes for conservation laws in a data-driven fashion. Furthermore, the learned policy network generalizes well to other situations such as different initial conditions, mesh sizes, temporal discrete schemes, etc. The paper ends with a conclusion in section 5, where possible future research directions are also discussed.

## 2 PRELIMINARIES

### 2.1 NOTATIONS

In this paper, we consider solving the following 1-D conservation laws:

$$u_t(x,t) + f_x(u(x,t)) = 0, \ \ a \le x \le b, \ t \in [0,T], \ \ u(x,0) = u_0(x). \tag{1}$$

For example, $f = \frac{u^2}{2}$ is the famous Burger's Equation. We discretize the $(x,t)$-plane by choosing a mesh with spatial size $\Delta x$ and temporal step size $\Delta t$, and define the discrete mesh points $(x_j, t_n)$ by

$$x_j = a + j\Delta x, \ t_n = n\Delta t \ \ \text{with } j = 0, 1, ..., J = \frac{b-a}{\Delta x}, \ n = 0, 1, ..., N = \frac{T}{\Delta t}.$$

We denote $x_{j+\frac{1}{2}} = x_j + \Delta x/2 = a + (j + \frac{1}{2})\Delta x$. The finite difference methods will produce approximations $U_j^n$ to the solution $u(x_j, t_n)$ on the given discrete mesh points. We denote point-wise values of the true solution to be $u_j^n = u(x_j, t_n)$, and the true point-wise flux values to be $f_j^n = f(u(x_j, t_n))$.

### 2.2 WENO – WEIGHTED ESSENTIALLY NON-OSCILLATORY SCHEMES

WENO (Weighted Essentially Non-Oscillatory) (Liu et al., 1994) is a family of high order accurate finite difference schemes for solving hyperbolic conservation laws, and has been successful for many practical problems. The key idea of WENO is a nonlinear adaptive procedure that automatically chooses the smoothest local stencil to reconstruct the numerical flux. Generally, a finite difference method solves Eq.1 by using a conservative approximation to the spatial derivative of the flux:

$$\frac{du_j(t)}{dt} = -\frac{1}{\Delta x}\left(\hat{f}_{j+\frac{1}{2}} - \hat{f}_{j-\frac{1}{2}}\right), \tag{2}$$

where $u_j(t)$ is the numerical approximation to the point value $u(x_j, t)$ and $\hat{f}_{j+\frac{1}{2}}$ is the numerical flux generated by a **numerical flux policy**

$$\hat{f}_{j+\frac{1}{2}} = \pi^f(u_{j-r}, ..., u_{j+s}),$$

which is manually designed. Note that the term "numerical flux policy" is a new terminology that we introduce in this paper, which is exactly the policy we shall learn using RL. In WENO, $\pi^f$ works as follows. Using the physical flux values $\{f_{j-2}, f_{j-1}, f_j\}$, we could obtain a $3^{th}$ order accurate polynomial interpolation $\hat{f}_{j+\frac{1}{2}}^{-2}$, where the indices $\{j-2, j-1, j\}$ is called a 'stencil'. We could also use the stencil $\{j-1, j, j+1\}$, $\{j, j+1, j+2\}$ or $\{j+1, j+2, j+3\}$ to obtain another three interpolants $\hat{f}_{j+\frac{1}{2}}^{-1}$, $\hat{f}_{j+\frac{1}{2}}^{0}$ and $\hat{f}_{j+\frac{1}{2}}^{1}$. The key idea of WENO is to average (with properly designed weights) all these interpolants to obtain the final reconstruction: $\hat{f}_{j+\frac{1}{2}} = \sum_{r=-2}^{1} w_r \hat{f}_{j+1/2}^r$, $\sum_{r=-2}^{1} w_r = 1$.

The weight $w_i$ depends on the smoothness of the stencil. A general principal is: the smoother is the stencil, the more accurate is the interpolant and hence the larger is the weight. To ensure convergence, we need the numerical scheme to be consistent and stable (LeVeque, 1992). It is known that WENO schemes as described above are consistent. For stability, upwinding is required in constructing the flux. The most easy way is to use the sign of the Roe speed $\bar{a}_{j+\frac{1}{2}} = (f_{j+\frac{1}{2}} - f_{j-\frac{1}{2}})/(u_{j+\frac{1}{2}} - u_{j-\frac{1}{2}})$ to determine the upwind direction: if $\bar{a}_{j+\frac{1}{2}} \ge 0$, we only average among the three interpolants $\hat{f}_{j+\frac{1}{2}}^{-2}$, $\hat{f}_{j+\frac{1}{2}}^{-1}$ and $\hat{f}_{j+\frac{1}{2}}^{0}$; if $\bar{a}_{j+\frac{1}{2}} < 0$, we use $\hat{f}_{j+\frac{1}{2}}^{-1}$, $\hat{f}_{j+\frac{1}{2}}^{0}$ and $\hat{f}_{j+\frac{1}{2}}^{1}$.

**Some further thoughts.** WENO achieves optimal order of accuracy (up to 5) at the smooth region of the solutions (Shu, 1998), while lower order of accuracy at singularities. The key of the WENO method lies in how to compute the weight vector $(w_1, w_2, w_3, w_4)$, which primarily depends on the smoothness of the solution at local stencils. In WENO, such smoothness is characterized by handcrafted formula, and was proven to be successful in many practical problems when coupled with high-order temporal discretization. However, it remains unknown whether there are better ways to combine the stencils so that optimal order of accuracy in smooth regions can be reserved while, at the

same time, higher accuracy can be achieved near singularities. Furthermore, estimating the upwind directions is another key component of WENO, which can get quite complicated in high-dimensional situations and requires lots of logical judgments (i.e. "if/else"). Can we ease the (some time painful) coding and improve the estimation at the aid of machine learning?

## 3 METHODS

In this section we present how to employ reinforcement learning to solve the conservation laws given by Eq.1. To better illustrate our idea, we first show in general how to formulate the process of numerically solving a conservation law into an MDP. We then discuss how to incorporate a policy network with the WENO scheme. Our policy network targets at the following two key aspects of WENO: **(1) Can we learn to choose better weights to combine the constructed fluxes? (2) Can we learn to automatically judge the upwind direction, without complicated logical judgments?**

### 3.1 MDP FORMULATION

---

**Algorithm 1:** A Conservation Law Solving Procedure

1 Input: initial values $u_0^0, u_1^0, ..., u_J^0$, flux $f(u)$, $\Delta x$, $\Delta t$, evolve time $N$, left shift $r$ and right shift $s$.
2 Output: $\{U_j^n | \, j = 0, ..., J, \; n = 1, ..., N\}$
3 $U_j^0 = u_j^0, \; j = 0, ..., J$
4 **for** $n = 1$ **to** $N$ **do**
5     **for** $j = 0$ **to** $J$ **do**
6        Compute the numerical flux $\hat{f}_{j-\frac{1}{2}}^n = \pi^f(U_{j-r-1}^{n-1}, U_{j-r}^{n-1}, ..., U_{j+s-1}^{n-1})$ and $\hat{f}_{j+\frac{1}{2}}^n = \pi^f(U_{j-r}^{n-1},$
          $U_{j-r+1}^{n-1}, ..., U_{j+s}^{n-1})$, e.g., using the WENO scheme
7        Compute $\frac{du_j(t)}{dt} = -\frac{1}{\Delta x}(\hat{f}_{j+\frac{1}{2}}^n - \hat{f}_{j-\frac{1}{2}}^n)$
8        Compute $U_j^n = \pi^t(U_j^{n-1}, \frac{du_j(t)}{dt})$, e.g., using the Euler scheme $U_j^n = U_j^{n-1} + \Delta t \frac{du_j(t)}{dt}$
9 **Return** $\{U_j^n | \, j = 0, ..., J, \; n = 1, ..., N\}$

---

As shown in Algorithm 1, the procedure of numerically solving a conservation law is naturally a sequential decision making problem. The key of the procedure is the numerical flux policy $\pi^f$ and the temporal scheme $\pi^t$ as shown in line 6 and 8 in Algorithm 1. Both policies could be learned using RL. However, in this paper, we mainly focus on using RL to learn the numerical flux policy $\pi^f$, while leaving the temporal scheme $\pi^t$ with traditional numerical schemes such as the Euler scheme or the Runge–Kutta methods. A quick review of RL is given in the appendix.

Now, we show how to formulate the above procedure as an MDP and the construction of the state $S$, action $A$, reward $r$ and transition dynamics $P$. Algorithm 2 shows in general how RL is incorporated into the procedure. In Algorithm 2, we use a single RL agent. Specifically, when computing $U_j^n$:

- The **state** for the RL agent is $s_j^n = g_s(U_{j-r-1}^{n-1}, ..., U_{j+s}^{n-1})$, where $g_s$ is the state function.

- In general, the **action** of the agent is used to determine how the numerical fluxes $\hat{f}_{j+\frac{1}{2}}^n$ and $\hat{f}_{j-\frac{1}{2}}^n$ is computed. In the next subsection, we detail how we incorporate $a_j^n$ to be the linear weights of the fluxes computed using different stencils in the WENO scheme.

- The **reward** should encourage the agent to generate a scheme that minimizes the error between its approximated value and the true value. Therefore, we define the reward function as $r_j^n = g_r(U_{j-r-1}^n - u_{j-r-1}^n, \cdots, U_{j+s}^n - u_{j+s}^n)$, e.g., a simplest choice is $g_r = -|| \cdot ||_2$.

- The **transition dynamics** $P$ is fully deterministic, and depends on the choice of the temporal scheme at line 10 in Algorithm 2. Note that the next state can only be constructed when we have obtained all the point values in the next time step, i.e., $s_j^{n+1} = g_s(U_{j-r-1}^n, ..., U_{j+s}^n)$ does not only depends on action $a_j^n$, but also on actions $a_{j-r-1}^n, ..., a_{j+s}^n$ (action $a_j^n$ can only determine the value $U_j^n$). This subtlety can be resolved by viewing the process under the framework of multi-agent RL, in which at each mesh point $j$ we use a distinct agent $A_j^{RL}$, and the next state $s_j^{n+1} = g_s(U_{j-r-1}^n, ..., U_{j+s}^n)$ depends on these agents' joint action $\mathbf{a_j^n} = (a_{j-r-1}^n, ..., a_{j+s}^n)$.

However, it is impractical to train $J$ different agents as $J$ is usually very large, therefore we enforce the agents at different mesh point $j$ to share the same weight, which reduces to case of using just a single agent. The single agent can be viewed as a counterpart of a human designer who decides on the choice of a *local scheme* based on the current state in traditional numerical methods.

---

**Algorithm 2:** General RL Running Procedure

1   Input: initial values $u_0^0, ..., u_J^0$, flux $f(u)$, $\Delta x$, $\Delta t$, evolve time $N$, left shift $r$, right shift $s$ and RL policy $\pi^{RL}$
2   Output: $\{U_j^n \mid j = 0, ..., J, \ n = 1, ..., N\}$
3   $U_j^0 = u_j^0, \ j = 0, ..., J$
4   **for** Many iterations **do**
5      Construct initial states $s_j^0 = g_s(U_{j-r-1}^0, ..., U_{j+s}^0)$ for $j = 0, ..., J$
6      **for** $n = 1$ to $N$ **do**
7          **for** $j = 0$ to $J$ **do**
8              Compute the action $a_j^n = \pi^{RL}(s_j^n)$ that determines how $\hat{f}_{j+\frac{1}{2}}^n$ and $\hat{f}_{j-\frac{1}{2}}^n$ is computed
9              Compute $\frac{du_j(t)}{dt} = -\frac{1}{\Delta x}(\hat{f}_{j+\frac{1}{2}}^n - \hat{f}_{j-\frac{1}{2}}^n)$
10              Compute $U_j^n = \pi^t(U_j^{n-1}, \frac{du_j(t)}{dt})$, e.g., the Euler scheme $U_j^n = U_j^{n-1} + \Delta t \frac{du_j(t)}{dt}$
11              Compute the reward $r_j^n = g_r(U_{j-r-1}^n - u_{j-r-1}^n, \cdots, U_{j+s}^n - u_{j+s}^n)$.
12          Construct the next states $s_j^{n+1} = g_s(u_{j-r-1}^n, ..., u_{j+s}^n)$ for $j = 0, ..., J$
13          Use any RL algorithm to train the RL policy $\pi^{RL}$ with the transitions $\{(s_j^n, a_j^n, r_j^n, s_j^{n+1})\}_{j=0}^J$.
14   **Return** the well-trained RL policy $\pi^{RL}$.

---

## 3.2   RL Empowered WENO

We now present how to transfer the actions of the RL policy to the weights of WENO fluxes. Instead of directly using $\pi^{RL}$ to generate the numerical flux, we use it to produce the weights of numerical fluxes computed using different stencils in WENO. Since the weights are part of the configurations of the WENO scheme, our design of action essentially makes the RL policy a meta-learner, and enables more stable learning and better generalization power than directly generating the fluxes.

Specifically, at point $x_j$ (here we drop the time superscript $n$ for simplicity), to compute the numerical flux $\hat{f}_{j-\frac{1}{2}}$ and $\hat{f}_{j+\frac{1}{2}}$, we first construct four fluxes $\{\hat{f}_{j-\frac{1}{2}}^i\}_{i=-2}^1$ and $\{\hat{f}_{j+\frac{1}{2}}^i\}_{i=-2}^1$ using four different stencils just as in WENO, and then use the RL policy $\pi^{RL}$ to generate the weights of these fluxes:

$$\pi^{RL}(s_j) = \left( w_{j-\frac{1}{2}}^{-2}, w_{j-\frac{1}{2}}^{-1}, w_{j-\frac{1}{2}}^0, w_{j-\frac{1}{2}}^1, w_{j+\frac{1}{2}}^{-2}, w_{j+\frac{1}{2}}^{-1}, w_{j+\frac{1}{2}}^0, w_{j+\frac{1}{2}}^1 \right).$$

The numerical flux is then constructed by averaging these fluxes: $\hat{f}_{j-\frac{1}{2}} = \sum_{i=-2}^1 w_{j-\frac{1}{2}}^i \hat{f}_{j-\frac{1}{2}}^i$, and $\hat{f}_{j+\frac{1}{2}} = \sum_{i=-2}^1 w_{j+\frac{1}{2}}^i \hat{f}_{j+\frac{1}{2}}^i$.

Note that the determination of upwind direction is automatically embedded in the RL policy since it generates four weights at once. For instance, when the roe speed $\bar{a}_{j+\frac{1}{2}} \geq 0$, we expect the $4^{th}$ weight $w_{j+\frac{1}{2}}^1 \approx 0$ and when $\bar{a}_{j+\frac{1}{2}} < 0$, we expect $w_{j+\frac{1}{2}}^{-2} \approx 0$. Note that the upwind direction can be very complicated in a system of equations or in the high-dimensional situations, and using the policy network to automatically embed such a process could save lots of efforts in algorithm design and implementation. Our numerical experiments show that $\pi^{RL}$ can indeed automatically determine upwind directions for 1D scalar cases. Although this does not mean that it works for systems and/or in high-dimensions, it shows the potential of the proposed framework and value for further studies.

## 4   Experiments

In this section, we describe training and testing of the proposed RL conservation law solver and compare it with WENO. More comparisons and discussions can be found in the appendix.

## 4.1 SETUP

In this subsection, we explain the general training setup. We train the RL policy network on the Burger's equation, whose flux is computed as $f(u) = \frac{1}{2}u^2$. In all the experiments, we set the left-shift $r = 2$ and the right shift $s = 3$. The state function $g_s(s_j) = g_s(U_{j-r-1}, ..., U_{j+s})$ will generate two vectors: $s^l = (f_{j-r-1}, ..., f_{j+s-1}, \bar{a}_{j-\frac{1}{2}})$, and $s^r = (f_{j-r}, ..., f_{j+s}, \bar{a}_{j+\frac{1}{2}})$ for computing $\hat{f}_{j-\frac{1}{2}}$ and $\hat{f}_{j+\frac{1}{2}}$ respectively. $s_l$ and $s_r$ will be passed into the same policy neural network $\pi_\theta^{RL}$ to produce the desired actions, as described in section 3.2. The reward function $g_r$ simply computes the infinity norm, i.e., $g_r(U_{j-r-1} - u_{j-r-1}, ..., U_{j+s} - u_{j+s}) = -||(U_{j-r-1} - u_{j-r-1}, ..., U_{j+s} - u_{j+s})||_\infty$.

The policy network $\pi_\theta^{RL}$ is a feed-forward Multi-layer Perceptron with 6 hidden layers, each has 64 neurons and use Relu (Goodfellow et al., 2016) as the activation function. We use the Deep Deterministic Policy Gradient Algorithm (Lillicrap et al., 2015) to train the RL policy.

To guarantee the generalization power of the trained RL agent, we randomly sampled 20 initial conditions in the form $u_0(x) = a + b \cdot \text{func}(c\pi x)$, where $|a| + |b| \leq 3.5$, $\text{func} \in \{sin, cos\}$ and $c \in \{2, 4, 6\}$. The goal of generating such kind of initial conditions is to ensure they have similar degree of smoothness and thus similar level of difficulty in learning. The computation domain is $-1 \leq x \leq 1$ and $0 \leq t \leq 0.8$ with $\Delta x = 0.02$, $\Delta t = 0.004$, and evolve steps $N = 200$ (which ensures the appearance of shocks). When training the RL agent, we use the Euler scheme for temporal discretization. The true solution needed for reward computing is generated using WENO on the same computation domain with $\Delta x = 0.001$, $\Delta t = 0.0002$ and the 4th order Runge-Kutta (RK4).

In the following, we denote the policy network that generates the weights of the WENO fluxes (as described in section 3.2) as RL-WENO. We randomly generated another different 10 initial conditions in the same form as training for testing.

| $\Delta t$ \ $\Delta x$ | 0.02 | | 0.04 | | 0.05 | |
|---|---|---|---|---|---|---|
| | RL-WENO | WENO | RL-WENO | WENO | RL-WENO | WENO |
| 0.002 | 5.66 (1.59) | 5.89 (1.74) | 8.76 (2.50) | 9.09 (2.62) | 9.71 (2.42) | 10.24 (2.84) |
| 0.003 | 5.64 (1.54) | 5.86 (1.67) | 8.73 (2.46) | 9.06 (2.58) | 9.75 (2.41) | 10.28 (2.81) |
| 0.004 | 5.63 (1.55) | 5.81 (1.66) | 8.72 (2.46) | 9.05 (2.55) | 9.61 (2.42) | 10.13 (2.84) |
| 0.005 | 5.08 (1.46) | 5.19 (1.58) | 8.29 (2.34) | 8.58 (2.47) | 9.30 (2.26) | 9.78 (2.69) |
| 0.006 | - | - | 8.71 (2.49) | 9.02 (2.61) | 9.72 (2.38) | 10.24 (2.80) |
| 0.007 | - | - | 8.56 (2.49) | 8.84 (2.62) | 9.59 (2.41) | 10.12 (2.80) |
| 0.008 | - | - | 8.68 (2.55) | 8.93 (2.66) | 9.57 (2.49) | 10.06 (2.92) |

Table 1: Comparison of relative errors ($\times 10^{-2}$) of RL-WENO and WENO with standard deviations of the errors among 10 trials in the parenthesis. Temporal discretization: RK4; flux function: $\frac{1}{2}u^2$.

| $\Delta t$ \ $\Delta x$ | 0.02 | | 0.04 | | 0.05 | |
|---|---|---|---|---|---|---|
| | RL-WENO | WENO | RL-WENO | WENO | RL-WENO | WENO |
| 0.002 | 4.85 (1.15) | 5.17 (1.26) | 7.77 (1.95) | 8.05 (2.02) | 8.16 (1.93) | 8.56 (2.19) |
| 0.003 | - | - | 7.79 (1.96) | 8.06 (2.03) | 8.18 (1.92) | 8.59 (2.18) |
| 0.004 | - | - | 7.72 (1.93) | 7.98 (2.01) | 8.15 (1.95) | 8.54 (2.20) |
| 0.005 | - | - | - | - | 8.18 (1.94) | 8.55 (2.15) |

Table 2: Comparison of relative errors ($\times 10^{-2}$) of RL-WENO and WENO with standard deviations of the errors among 10 trials in the parenthesis. Temporal discretization: RK4; flux function: $\frac{1}{16}u^4$.

## 4.2 RESULTS

We compare the performance of RL-WENO and WENO. We also test whether the trained RL policy can generalize to different temporal discretization schemes, mesh sizes and flux functions that are not included in training. Table 1 and Table 2 present the comparison results, where the number shows the relative error (computed as $\frac{||U-u||_2}{||u||_2}$ with the 2-norm taking over all $x$) between the approximated solution $U$ and the true solution $u$, averaged over 250 evolving steps ($T = 1.0$) and 10 random initial values. Numbers in the bracket shows the standard deviation over the 10 initial conditions. Several entries in the table are marked as '-' because the corresponding CFL number is not small enough

to guarantee convergence. Recall that training of the RL-WENO was conducted with Euler time discretization, $(\Delta x, \Delta t) = (0.02, 0.004)$, $T = 0.8$ and $f(u) = \frac{1}{2}u^2$.

Our experimental results show that, compared with the high order accurate WENO (5th order accurate in space and 4th order accurate in time), the linear weights learned by RL not only achieves smaller errors, but also generalizes well to: 1) longer evolving time ($T = 0.8$ for training and $T = 1.0$ for testing); 2) new time discretization schemes (trained on Euler, tested on RK4); 3) new mesh sizes (see Table 1 and Table 2 for results of varied $\Delta x$ and $\Delta t$); and 4) a new flux function (trained on $f(u) = \frac{1}{2}u^2$ shown in Table 1, tested on $\frac{1}{16}u^4$ Table 2).

Figure 1 shows some examples of the solutions. As one can see, the solutions generated by RL-WENO not only achieve the same accuracy as WENO at smooth regions, but also have clear advantage over WENO near singularities which is particularly challenging for numerical PDE solvers and important in applications. Figure 2 shows that the learned numerical flux policy can indeed correctly determine upwind directions and generate local numerical schemes in an adaptive fashion. More interestingly, Figure 2 further shows that comparing to WENO, RL-WENO seems to be able to select stencils in a different way from it, and eventually leads to a more accurate solution. This shows that the proposed RL framework has the potential to surpass human experts in designing numerical schemes for conservation laws.

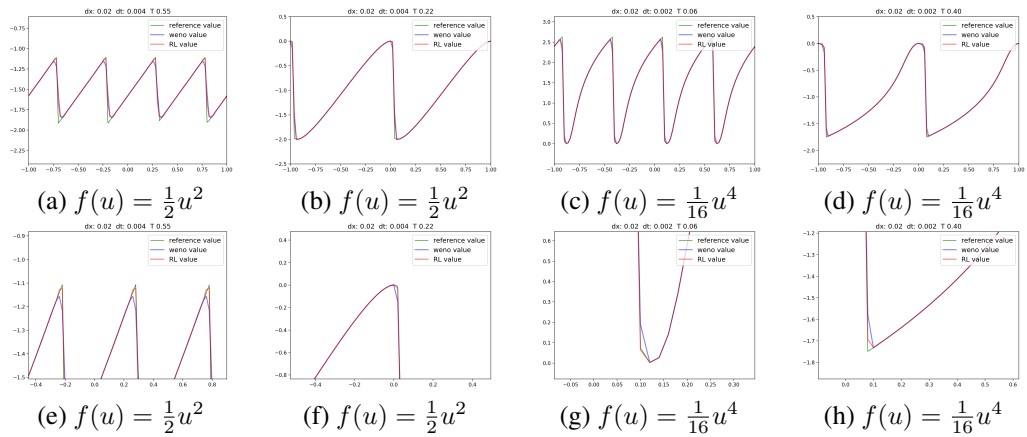

(a) $f(u) = \frac{1}{2}u^2$  (b) $f(u) = \frac{1}{2}u^2$  (c) $f(u) = \frac{1}{16}u^4$  (d) $f(u) = \frac{1}{16}u^4$

(e) $f(u) = \frac{1}{2}u^2$  (f) $f(u) = \frac{1}{2}u^2$  (g) $f(u) = \frac{1}{16}u^4$  (h) $f(u) = \frac{1}{16}u^4$

Figure 1: First row: solutions of RL-WENO (red), WENO (blue) and exact solutions (green). Second row: zoom-in views corresponding to the first row.

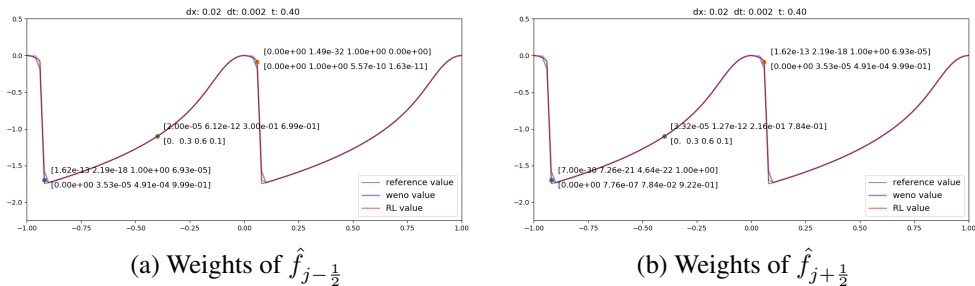

(a) Weights of $\hat{f}_{j-\frac{1}{2}}$  (b) Weights of $\hat{f}_{j+\frac{1}{2}}$

Figure 2: This figure compares the weights generated by the learned numerical flux policy $\pi^{RL}$ and those of WENO. The weights shown in (a) are $\{w^r_{j-\frac{1}{2}}\}^1_{r=-2}$; while those in (b) are $\{w^r_{j+\frac{1}{2}}\}^1_{r=-2}$. In each of the two plots, the 4 numbers in the upper bracket of each location are the weights of RL-WENO and those in the lower bracket are the weights of WENO. The relative errors of RL-WENO and WENO are $8.0 \times 10^{-3}$ and $2.5 \times 10^{-2}$ respectively.

## 5   CONCLUSION

In this paper, we proposed a general framework to learn how to solve 1-dimensional conservation laws via deep reinforcement learning. We first discussed how the procedure of numerically solving conservation laws can be naturally cast in the form of Markov Decision Process. We then elaborated how to relate notions in numerical schemes of PDEs with those of reinforcement learning. In particular, we introduced a numerical flux policy which was able to decide on how numerical flux should be designed locally based on the current state of the solution. We carefully design the action of our RL policy to make it a meta-learner. Our numerical experiments showed that the proposed RL based solver was able to outperform high order WENO and was well generalized in various cases.

As part of the future works, we would like to consider using the numerical flux policy to inference more complicated numerical fluxes with guaranteed consistency and stability. Furthermore, we can use the proposed framework to learn a policy that can generate adaptive grids and the associated numerical schemes. Lastly, we would like consider system of conservation laws in 2nd and 3rd dimensional space.

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

# A    COMPLEMENTARY EXPERIMENTS

## A.1    COMPARISON WITH SUPERVISED LEARNING (SL) BASED METHODS

We first note that most of the neural network based numerical PDE solvers cited in the introduction requires retraining when the initialization, terminal time, or the form of the PDE is changed; while the proposed RL solver is much less restricted as shown in our numerical experiments. This makes proper comparisons between existing NN-based solvers and our proposed solver very difficult. Therefore, to demonstrate the advantage of our proposed RL PDE solver, we would like to propose a new SL method that does not require retraining when the test setting (e.g. initialization, flux function, etc.) is different from the training.

However, as far as we are concerned, it is challenging to design such SL methods without formulating the problem into an MDP. One may think that we can use WENO to generate the weights for the stencil at a particular grid point on a dense grid, and use the weights of WENO generated from the dense grid as the label to train a neural network in the coarse grid. But such setting has a fatal flaw in that the stencils computed in the dense grids are very different from those in the coarse grids, especially near singularities. Therefore, good weights on dense grids might perform very poorly on coarse grids. In other words, simple imitation of WENO on dense grids is not a good idea. One might also argue that instead of learning the weights of the stencils, we could instead generate the discrete operators, such as the spatial discretization of $\frac{\partial u_j}{\partial x}$, or the temporal discretization of $\frac{\partial u_j}{\partial t}$, the numerical fluxes $f_{j+\frac{1}{2}}(u), f_{j-\frac{1}{2}}(u)$, etc., on a dense grid, and then use them as labels to train a neural network in the supervised fashion on a coarse grid. However, the major problem with such design is that there is no guarantee that the learned discrete operators obey the conservation property of the equations, and thus they may also generalize very poorly.

After formulating the problem into a MDP, there is indeed one way that we can use back-propagation (BP) instead of RL algorithms to optimize the policy network. Because all the computations on using the stencils to calculate the next-step approximations are differentiable, we can indeed use SL to train the weights. One possible way is to minimize the error (e.g. 2 norm) between the approximated and the true values, where the true value is pre-computed using a more accurate discretization on a fine mesh. The framework to train the SL network is described in Algorithm 3. Note that the framework to train the SL network is essentially the same as that of the proposed RL-WENO (Algorithm 2). The only difference is that we train the SL network using BP and the RL network using DDPG.

---

**Algorithm 3:** Using BP instead of RL algorithm to train the policy

1   Input: initial values $u_0^0, ..., u_J^0$, flux $f(u)$, $\Delta x$, $\Delta t$, evolve time $N$, left shift $r$, right shift $s$ and a neural network $\pi^\theta$

2   Output: $\{U_j^n \,|\, j = 0, ..., J, \; n = 1, ..., N\}$

3   $U_j^0 = u_j^0, \; j = 0, ..., J$

4   **for** Many iterations **do**

5      Construct initial states $s_j^0 = g_s(U_{j-r-1}^0, ..., U_{j+s}^0)$ for $j = 0, ..., J$

6      **for** $n = 1$ **to** $N$ **do**

7          **for** $j = 0$ **to** $J$ **do**

8              Compute the weights $(w_{j-\frac{1}{2}}^{n,-2}, w_{j-\frac{1}{2}}^{n,-1}, w_{j-\frac{1}{2}}^{n,0}, w_{j-\frac{1}{2}}^{n,1}, w_{j+\frac{1}{2}}^{n,-2}, w_{j+\frac{1}{2}}^{n,-1}, w_{j+\frac{1}{2}}^{n,0}, w_{j+\frac{1}{2}}^{n,1}) = \pi^\theta(s_j^n)$

9              Compute the fluxes $\hat{f}_{j-\frac{1}{2}}^n = \sum_{i=-2}^1 w_{j-\frac{1}{2}}^{n,i} \hat{f}_{j-\frac{1}{2}}^{n,i}$, $\hat{f}_{j+\frac{1}{2}}^n = \sum_{i=-2}^1 w_{j+\frac{1}{2}}^i \hat{f}_{j+\frac{1}{2}}^{n,i}$, where $\hat{f}_{j\pm\frac{1}{2}}^{n,i}$ are the fluxes computed by WENO

10              Compute $\frac{du_j(t)}{dt} = -\frac{1}{\Delta x}(\hat{f}_{j+\frac{1}{2}}^n - \hat{f}_{j-\frac{1}{2}}^n)$

11              Compute $U_j^n = \pi^t(U_j^{n-1}, \frac{du_j(t)}{dt})$, e.g., the Euler scheme $U_j^n = U_j^{n-1} + \Delta t \frac{du_j(t)}{dt}$

12              Compute the loss for $\theta$:
             $L_j^n(\theta) = ||(U_{j-r-1}^n - u_{j-r-1}^n, \cdots, U_{j+s}^n - u_{j+s}^n) - (U_{j-r-1}^n - u_{j-r-1}^n, \cdots, U_{j+s}^n - u_{j+s}^n)||_2^2$.

13              Perform a gradient descent on $\theta$ w.r.t $L_j^n(\theta)$

14          Construct the next states $s_j^{n+1} = g_s(u_{j-r-1}^n, ..., u_{j+s}^n)$ for $j = 0, ..., J$

15   **Return** the BP optimized policy $\pi^\theta$.

---

However, we argue that the main drawback of using SL (BP) to optimize the stencils in such a way is that it cannot enforce long-term accuracy and thus cannot outperform the proposed RL-WENO. To support such claims, we have added experiments using SL to train the weights of the stencils, and the results are shown in table 3 and 4. The SL policy is trained till it achieves very low loss (i.e., converges) in the training setting. However, as shown in the table, the SL-trained policy does not perform well overall. To improve longer time stability, one may argue that we could design the loss of SL to be the accumulated loss over multiple prediction steps, but in practice as the dynamics of our problem (computations for obtaining multiple step approximations) is highly non-linear, thus the gradient flow through multiple steps can be highly numerically unstable, making it difficult to obtain a decent result.

| $\Delta t$ \ $\Delta x$ | 0.02 | | | 0.04 | | | 0.05 | | |
|---|---|---|---|---|---|---|---|---|---|
| | RL-WENO | SL | WENO | RL-WENO | SL | WENO | RL-WENO | SL | WENO |
| 0.002 | 5.66 (1.59) | 7.86 (1.23) | 5.89 (1.74) | 8.76 (2.50) | 12.48 (0.78) | 9.09 (2.62) | 9.71 (2.42) | 12.14 (0.44) | 10.24 (2.84) |
| 0.003 | 5.64 (1.54) | 7.77 (1.26) | 5.86 (1.67) | 8.73 (2.46) | 12.44 (0.78) | 9.06 (2.58) | 9.75 (2.41) | 12.13 (0.41) | 10.28 (2.81) |
| 0.004 | 5.63 (1.55) | 7.72 (1.14) | 5.81 (1.66) | 8.72 (2.46) | 12.44 (0.64) | 9.05 (2.55) | 9.61 (2.42) | 12.14 (0.45) | 10.13 (2.84) |
| 0.005 | 5.08 (1.46) | 7.14 (1.37) | 5.19 (1.58) | 8.29 (2.34) | 12.06 (0.86) | 8.58 (2.47) | 9.30 (2.26) | 11.86 (0.38) | 9.78 (2.69) |
| 0.006 | - | - | - | 8.71 (2.49) | 12.33 (0.73) | 9.02 (2.61) | 9.72 (2.38) | 12.14 (0.41) | 10.24 (2.80) |
| 0.007 | - | - | - | 8.56 (2.49) | 12.29 (0.83) | 8.84 (2.62) | 9.59 (2.41) | 12.06 (0.45) | 10.12 (2.80) |
| 0.008 | - | - | - | 8.68 (2.55) | 12.22 (0.70) | 8.93 (2.66) | 9.57 (2.49) | 12.08 (0.46) | 10.06 (2.92) |

Table 3: Comparison of relative errors ($\times 10^{-2}$) of RL-WENO, WENO, and SL-trained policy with standard deviations of the errors among 10 trials in the parenthesis. Temporal discretization: RK4; flux function: $\frac{1}{2}u^2$. RL-weno consistently outperforms WENO and SL-trained policy in all test cases.

| $\Delta t$ \ $\Delta x$ | 0.02 | | | 0.04 | | | 0.05 | | |
|---|---|---|---|---|---|---|---|---|---|
| | RL-WENO | SL | WENO | RL-WENO | SL | WENO | RL-WENO | SL | WENO |
| 0.002 | 4.85 (1.15) | 5.84 (0.79) | 5.17 (1.26) | 7.77 (1.95) | 8.60 (1.12) | 8.05 (2.02) | 8.16 (1.93) | 8.42 (1.00) | 8.56 (2.19) |
| 0.003 | - | - | - | 7.79 (1.96) | 8.62 (1.12) | 8.06 (2.03) | 7.70 (1.96) | 8.42 (0.98) | 8.59 (2.18) |
| 0.004 | - | - | - | 7.72 (1.93) | 8.55 (1.15) | 7.98 (2.01) | 8.15 (1.95) | 8.41 (1.02) | 8.54 (2.20) |
| 0.005 | - | - | - | - | - | - | 8.18 (1.94) | 8.40 (1.03) | 8.55 (2.15) |

Table 4: Comparison of relative errors ($\times 10^{-2}$) of RL-WENO, WENO, and SL-trained policy with standard deviations of the errors among 10 trials in the parenthesis. Temporal discretization: RK4; flux function: $\frac{1}{2}u^4$. RL-weno consistently outperforms WENO and SL-trained policy in all test cases.

## A.2 RL-WENO'S PERFORMANCE ON SMOOTH AND SINGULAR REGIONS

As mentioned in section 2.2, WENO itself already achieves an optimal order of accuracy in the smooth regions. Since RL-WENO can further improve upon WENO, it must have obtained higher accuracy especially near singularities. Here we provide additional demonstrations on how RL-WENO performs in the smooth/singular regions. We run RL-WENO and WENO on a set of initial conditions, and record the approximation errors at every locations and then separate the errors in the smooth and singular regions for every time step. We then compute the distribution of the errors on the entire spatial-temporal grids with multiple initial conditions. The results are shown in figure 3. In figure 3, the $x$-axis is the logarithmic (base 10) value of the error and the y-axis is the number of grid points whose error is less than the corresponding value on the $x$-axis, i.e., the accumulated distribution of the errors. The results show that RL-WENO indeed performs better than WENO near singularities. RL-WENO even achieves better accuracy than WENO in the smooth region when the flux function is $\frac{1}{16}u^4$.

## A.3 INFERENCE TIME OF RL-WENO AND WENO

In this subsection we report the inference time of RL-WENO and WENO. Although the computation complexity of the trained RL policy (a MLP) is higher than that of WENO, we could parallel and accelerate the computations using GPU.

Our test is conducted in the following way: for each grid size $\Delta x$, we fix the initial condition as $u_0(x) = 1 + cos(6\pi x)$, the evolving time $T = 0.8$ and the flux function $f = u^2$. We then use RL-WENO and WENO to solve the problem 20 times, and report the average running time. For completeness, we also report the relative error of RL-WENO and WENO in each of these grid sizes in table 6. Note that the relative error is computed on average of several initial functions, and our RL-WENO policy is only trained on grid $(\Delta x, \Delta t) = (0.02, 0.004)$.

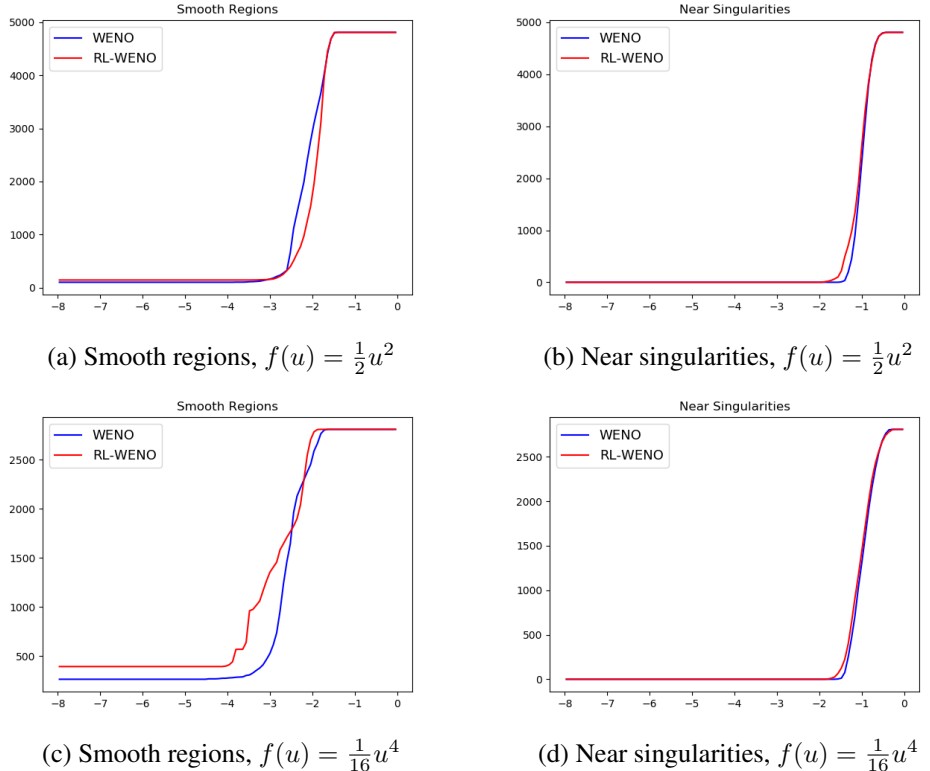

Figure 3: These figures show the total number of grids whose error is under a specific value (i.e. the accumulated distribution function). The $x$-axis is the error in logarithmic (base 10) scale. (a) and (c) show the distribution in smooth regions, (b) and (d) are near singularities.

For RL-WENO, we test it on both CPU and on GPU; For WENO, we test it purely on CPU, with a well-optimized version (e.g., good numpy vectorization in python), and a poor-implemented version (e.g., no vectorization, lots of loops). The CPU used for the tests is a custom Intel CORE i7, and the GPU is a custom NVIDIA GTX 1080. The results are shown in table 5.

| $(\Delta x, \Delta t)$ | RL-WENO(CPU) | RL-WENO(GPU) | WENO-optimized | WENO-poor |
|---|---|---|---|---|
| (0.02, 0.004) | 2.490 | 1.650 | **0.148** | 2.739 |
| (0.01, 0.002) | 7.720 | 1.700 | **0.349** | 10.778 |
| (0.005, 0.001) | 26.70 | 1.628 | **0.921** | 44.23 |
| (0.002, 0.0004) | 110.92 | **1.611** | 1.961 | 277.88 |

Table 5: Average inference time (in seconds) for RL-WENO and WENO. Bold numbers are the smallest ones.

| $(\Delta x, \Delta t)$ | RL-WENO error | WENO error |
|---|---|---|
| (0.02, 0.004) | 3.73(0.40) | 4.08(0.23) |
| (0.01, 0.002) | 1.86(0.17) | 1.99(0.12) |
| (0.005, 0.001) | 1.00(0.05) | 0.93(0.01) |
| (0.002, 0.0004) | 0.48(0.03) | 0.39(0.02) |

Table 6: Relative error of RL-WENO and WENO ($\times 10^{-2}$) on grid sizes tested in table 5. Note RL-WENO is only trained on grid $(\Delta x, \Delta t) = (0.02, 0.004)$

From the table we can tell that as $\Delta x$ decreases, i.e., as the grid becomes denser, all methods, except for the RL-WENO (GPU), requires significant more time to finish the computation. The reason that the time cost of the GPU-version of RL-WENO does not grow is that on GPU, we can compute

all approximations in the next step (i.e., to compute $(U_0^{t+1}, U_1^{t+1}, ..., U_J^{t+1})$ given $(U_0^t, U_1^t, ..., U_J^t)$, which dominates the computation cost of the algorithm) together in parallel. Thus, the increase of grids does not affect much of the computation time. Therefore, for coarse grid, well-optimized WENO indeed has clear speed advantage over RL-WENO (even on GPU), but on a much denser grid, RL-WENO (GPU) can be faster than well-optimized WENO by leveraging the paralleling nature of the algorithm.

## B  REVIEW OF REINFORCEMENT LEARNING

### B.1  REINFORCEMENT LEARNING

Reinforcement Learning (RL) is a general framework for solving sequential decision making problems. Recently, combined with deep neural networks, RL has achieved great success in various tasks such as playing video games from raw screen inputs (Mnih et al., 2015), playing Go (Silver et al., 2016), and robotics control (Schulman et al., 2017). The sequential decision making problem RL tackles is usually formulated as a Markov Decision Process (MDP), which comprises five elements: the state space $S$, the action space $A$, the reward $r : S \times A \to \mathcal{R}$, the transition probability of the environment $P : S \times A \times S \to [0, 1]$, and the discounting factor $\gamma$. The interactions between an RL agent and the environment forms a trajectory $\tau = (s_0, a_0, r_0, ..., s_T, a_T, r_T, ...)$. The return of $\tau$ is the discounted sum of all its future rewards:

$$G(\tau) = \sum_{t=0}^{\infty} \gamma^t r_t$$

Similarly, the return of a state-action pair $(s_t, a_t)$ is:

$$G(s_t, a_t) = \sum_{l=t}^{\infty} \gamma^{l-t} r_l$$

A policy $\pi$ in RL is a probability distribution on the action $A$ given a state $S$: $\pi : S \times A \to [0, 1]$. We say a trajectory $\tau$ is generated under policy $\pi$ if all the actions along the trajectory is chosen following $\pi$, i.e., $\tau \sim \pi$ means $a_t \sim \pi(\cdot|s_t)$ and $s_{t+1} \sim P(\cdot|s_t, a_t)$. Given a policy $\pi$, the value of a state $s$ is defined as the expected return of all the trajectories when the agent starts at $s$ and then follows $\pi$:

$$V^\pi(s) = E_\tau[G(\tau)|\tau(s_0) = s, \tau \sim \pi]$$

Similarly, the value of a state-action pair is defined as the expected return of all trajectories when the agent starts at $s$, takes action $a$, and then follows $\pi$:

$$Q^\pi(s, a) = E_\tau[G(\tau)|\tau(s_0) = s, \tau(a_0) = a, \tau \sim \pi]$$

As aforementioned in introduction, in most RL algorithms the policy $\pi$ is optimized with regards to the values $Q^\pi(s, a)$, thus naturally guarantees the long-term accumulated rewards (in our setting, the long-term accuracy of the learned schemes). Bellman Equation, one of the most important equations in RL, connects the value of a state and the value of its successor state:

$$Q^\pi(s, a) = r(s, a) + \gamma E_{s' \sim P(\cdot|s,a), a' \sim \pi(\cdot|s')}[Q^\pi(s', a')]$$
$$V^\pi(s) = E_{a \sim \pi(\cdot|s), s' \sim P(\cdot|s', a)}[r(s, a) + \gamma V^\pi(s')]$$

The goal of RL is to find a policy $\pi$ to maximize the expected discounted sum of rewards starting from the initial state $s_0$, $J(\pi) = E_{s_0 \sim \rho}[V^\pi(s_0)]$, where $\rho$ is the initial state distribution. If we parameterize $\pi$ using $\theta$, then we can optimize it using the famous policy gradient theorem:

$$\frac{dJ(\pi_\theta)}{d\theta} = E_{s \sim \rho^{\pi_\theta}, a \sim \pi_\theta}[\nabla_\theta \log \pi_\theta(a|s) Q^{\pi_\theta}(s, a)]$$

where $\rho^{\pi_\theta}$ is the state distribution deduced by the policy $\pi_\theta$. In this paper we focus on the case where the action space $A$ is continuous, and a lot of mature algorithms has been proposed for such a case, e.g., the Deep Deterministic Policy Gradient (DDPG) (Lillicrap et al., 2015), the Trust Region Policy Optimization algorithm (Schulman et al., 2015), and etc.

