# OpenReview forum: "Learning to Discretize: Solving 1D Scalar Conservation Laws via Deep Reinforcement Learning"
_ICLR.cc/2020/Conference — Reject_

### Official Review · AnonReviewer2 · 2019-10-19
**Official Blind Review #2**

**Rating:** 3

**Review:**


##### Rebuttal Response:
The other reviewers seem to have understood more than me. Their opinion and the rebuttal did not convince me to update my score. In my opinion the writing must be adapted to be interesting to the ICLR community and the bigger picture should be highlighted more, as the bigger picture is remains quite unclear at the current state.


##### Review:
Summary:
[...]


Conclusion:
I have read the paper multiple times and I still have a problem summarizing the paper with my own words. The contributions summarize the most fundamental works of RL but do not really relate these methods to the proposed approach. Therefore, I am still uncertain about the general motivation and intention of the work as well as the evaluation. Currently I vote for borderline reject as I am familiar with RL & PDE'S but do not understand the motivation and intention. I am leaning towards rejection as the paper is a resubmission from Neurips and has not been substantially improved. However, I am not certain about my evaluation. I am happy to adapt my vote based on the other reviewers and a clarified and better structured paper, which can be submitted during the rebuttal.

**Experience Assessment:**

I have published one or two papers in this area.

**Review Assessment: Checking Correctness Of Derivations And Theory:**

N/A

**Review Assessment: Checking Correctness Of Experiments:**

N/A

**Review Assessment: Thoroughness In Paper Reading:**

I read the paper at least twice and used my best judgement in assessing the paper.

---

> ### Author Response · Authors · 2019-11-15
> **Response**
>
> Thanks the reviewer for your feedbacks.
>
> We have updated the manuscript for the following 3 parts, where all the updates are in the "Complementary Experiments" section (section A) in the Appendix. 1) We added experiments on comparing our RL-based method and a SL-based method, in appendix A.1. 2) We add more figures analyzing the performance of our RL policy on smooth regions and near singularities of the PDE solutions in appendix A.2. 3) We report and compare the inference time of our RL policy and WENO in appendix A.3. Currently we put these contents in appendix, but if the paper gets accepted, we would then incorporate them into the main body in the final version.

---

### Official Review · AnonReviewer1 · 2019-10-21
**Official Blind Review #1**

**Rating:** 3

**Review:**

In this paper, the author maps the problem of time series PDE into a naive reinforcement learning problem. Under the MDP assumption, the author sets the initial state of the particles as the current state, the flux at all spaces as the possible actions, and map the state-action pair deterministically to the next state of the particle diffusion. The reward is defined as the two norms between the prediction and the Burger’s equation. The naiveness comes from the fact that the typical reinforcement learning problem, the agent needs to decide how to choose an action. In this paper, it is formulated as an intrinsic proper that follows Burger’s equation instead.

While the motivation is interesting, the author argues this work is novel due to it does not fall under supervised learning, but rather reinforcement learning. This perspective is not completely correct. The correct category for this work would be more similar to imitation learning using WANO’s algorithm as the expert label. This is a field of supervised reinforcement learning.

The author’s work has brought the possibility of using neural network architecture in the field of particle diffusion. The benefit is the improved estimation of how particles diffuse in long-horizon conditions. The author has shown in their paper their simple fully connected network has already performed better prediction than the current state of the art non-neural network model: WENO.

While the framing of the problem is perhaps novel in the space of PDE, algorithmically there needs to have a breakthrough or new invention. The lack of comparison with other neural-network-based models also hurts the credibility of the model. Therefore, I reject this paper under the ICLR conference. I would suggest that this paper would be better suited as a paper submission under the perspective science field conference instead.

Some suggestions to further improve this paper: The author could add CNN and RNN structure to the prediction model. These structures would further expand other possibilities in the solution space. CNN would help turn the limited 1D problem to a higher-dimensional, a more real-world like problem space. RNN is known for its’ ability to model long horizon problems, perhaps even better breakthrough would happen with these architectures.

As a whole, the paper is written very well such that even nonexpert can grab onto the logic flow of this paper. The weaknesses of the paper are the lack of diversity in comparison with other models and the paper needs some level of novel breakthrough in an algorithmic sense.


**Experience Assessment:**

I have read many papers in this area.

**Review Assessment: Checking Correctness Of Derivations And Theory:**

I assessed the sensibility of the derivations and theory.

**Review Assessment: Checking Correctness Of Experiments:**

I assessed the sensibility of the experiments.

**Review Assessment: Thoroughness In Paper Reading:**

I read the paper at least twice and used my best judgement in assessing the paper.

---

> ### Author Response · Authors · 2019-11-15
> **Response**
>
> We thank the reviewer for the constructive feedbacks. We answer the reviewer's major concerns as below.
>
> -- manuscirpt update
> We have updated the manuscript for the following 3 parts, where all the updates are in the "Complementary Experiments" section (section A) in the Appendix. 1) We added experiments on comparing our RL-based method and a SL-based method, in appendix A.1. 2) We add more figures analyzing the performance of our RL policy on smooth regions and near singularities of the PDE solutions in appendix A.2. 3) We report and compare the inference time of our RL policy and WENO in appendix A.3. Currently we put these contents in appendix, but if the paper gets accepted, we would then incorporate them into the main body in the final version.
>
>
> -- "the mapping of the problem of time series PDE to a reinforcement learning problem is naive; algorithmically there needs to have a breakthrough or new invention"
> The reviewer mentioned that our casting of the problem of solving a evolutionary PDE to a RL problem is naive due to our design of the agent's action. We want to clarify that there are actually a lot of subtleties in such seemingly easy mapping, which we could not explain in detail in the paper due to the page limit. For example, there are numerous different designs for the agent's action: we could have designed it to be a spatial discretization $\frac{\partial u_j}{\partial x}$, a temporal discretization $\frac{\partial u_j}{\partial t}$, or the flux $f_{j+\frac12}, f_{j-\frac12}$. We had initial experiments with such designs, but the main problem is that they either do not obey the conservation property of the equation (when you directly learn a spatial or temporal discretization), or they generalize poorly (when you directly learn the fluxes). After countless trial and error by ourselves, we found that the current setting works the best and indeed improves over WENO near singularities.
>
> The reviewer also mentioned that our methods lack an algorithmeically invention. We admit that we were just using standard RL algorithm for training. However, part of our innovation comes from the formulation of the problem to a proper MDP. One of them is explained in the last bullet of section 3.1 in the paper, and we rephrase here: ``since the next state depends on not a single but several actions at the current step, the formulated MDP is essentially a multi-agent RL problem. However, it is impractical to train a number of individual agents that is equal to the grid number, so we share the weight among the agents, and the problem can be addressed under a single agent view".  This design further leads to another invention of our formulation: with a single shared RL policy operating on the current 1-D line of grid points sequentially from "left" to "right" (i.e., $\pi^{RL}$ operates on $(U_0, ..., U_{r+s})$, then $(U_1, ..., U_{r+s+1})$, till $(U_{J-r-s+1}, ... U_J)$ ), we are essentially applying a special convolution to the current line of grid points to generate the stencil, where the kernel is the non-linear MLP RL policy. We will update these discussions in the paper if accepted.
>
>
> -- "the method is using weno as label. the method is actually supervised reinforcement learning"
> The reviewer mentioned that we are doing "supervised reinforcement learning", or imitation learning using WENO as the label. We want to clarify that we are not imitating WENO. Instead of learning towards WENO's output, we are minimizing the error between the RL generated solution and the true solution which is obtained from WENO scheme on a much denser grid. One could use any other algorithm than WENO to generate the true solution. Besides, figure 2 in our paper demonstrates that our RL policy has learned to generate very different stencils from WENO, which further proves the learned RL policy is not just imitating WENO. It attempts to surpass WENO by learning from the true solution.
>
>
> --"the lack of comparison with other neural-network-based models also hurts the credibility of the model."
> We have added more discussions on using other NN-based methods to learn the stencil, and also experiments of using a SL-based (actually, BP-based) method to train a NN to choose the stencil, in appendix A.1 of the revised manuscript. We found our RL trained NN consistently outperforms the SL trained NN, which verifies RL's advantage on guaranteeing long-term accuracy and generalization ability.
>
>
> --"using CNN and RNN as the policy network."
> We thank the reviewer's suggestion of using CNN and RNN for our RL policy. We agree that they have stronger representation power and will consider using them in future works.

---

### Official Review · AnonReviewer3 · 2019-10-23
**Official Blind Review #3**

**Rating:** 6

**Review:**

This paper proposes to use reinforcement learning for constructing discretziation stencils of numerical schemes. More specifically, the method focuses on the widely used WENO schemes, which are an established class of finite difference schemes. Within this context, the method aims for training models to infer the weighting for a specific stencil with eight flux terms.

For RL this task requires a continuous action space, and the DDPG algorithm is used for training the policy. The network itself is an MLP with 6 layers, and ca. 20000 weights in total. This is a significant number, given the focus on 1D problems.

The tests are quite thorough and interesting, while at the same time being limited in scope. The paper targets 1D cases, which make the problem very low-dimensional. Despite the simplicity, only a single data set (Burgers) is used, and a single modified target function with a u^4 term. Targeting 1D casesl, I would have expected a broader range of tests and model equations.

Despite the limited scope of the models, table 1 and 2 assess a nice range of different timestep and discretization parameters. I found it very interesting to see that the method consistently outperforms the regular WENO scheme. The gains are relatively small, with 4-5%, but WENO already represents a quite accurate scheme, so it's surely not easy to outperform it.

While reading the paper, I was wondering about the bigger picture, i.e. using RL in the context of discretization stencils. We have model equations, and discretized versions of all operators involved in training. Why employ a "brute force" approach like RL here? Wouldn't it be better in terms of efficiency and potentially also accuracy to train the stencils in a supervised manner, e.g., with a more accurate discretization as reference? One could argue that it would be expensive to pre-compute such data, but I think RL scales even worse to higher dimensional problems.

What's also missing in the current version is a more thorough discussion of inference and training performance. I guess that despite the small model problems, the training takes a substantial amount of time. And due to the large size of the trained model, which has to be evaluated for every single node in the 1D mesh, it's probably also quite slow. I think this is worth a discussion in the text. One could even estimate the number of operations necessary to evaluate the model, and run a higher-order WENO scheme for a "fair" comparison.

Minor, but in equation (1), I guess the t subscript should indicate a material derivative, and just just a time derivative, right? This could be clarified in the text (or written out).

I am somewhat on the edge with this paper - the 1D case for the two equations is carefully evaluated in the submission, and it's great to see the trained model can improve the accuracy across a fairly wide range of settings. As such, it's definitely a good and interesting first step. On the other hand, there are a range of open questions, as outlined above, and it's not clear whether the approach could be easily translated to higher dimensions. I hope the authors can clarify some of these points in the rebuttal, right now I'm leaning towards the positive side.


**Experience Assessment:**

I have published one or two papers in this area.

**Review Assessment: Checking Correctness Of Derivations And Theory:**

I assessed the sensibility of the derivations and theory.

**Review Assessment: Checking Correctness Of Experiments:**

I carefully checked the experiments.

**Review Assessment: Thoroughness In Paper Reading:**

I read the paper thoroughly.

---

> ### Author Response · Authors · 2019-11-15
> **Response**
>
> Thanks the reviewer for the useful feedbacks! Below are our responses.
>
> -- manuscirpt update
> We have updated the manuscript for the following 3 parts, where all the updates are in the "Complementary Experiments" section (section A) in the Appendix. 1) We added experiments on comparing our RL-based method and a SL-based method, in appendix A.1. 2) We add more figures analyzing the performance of our RL policy on smooth regions and near singularities of the PDE solutions in appendix A.2. 3) We report and compare the inference time of our RL policy and WENO in appendix A.3. Currently we put these contents in appendix, but if the paper gets accepted, we would then incorporate them into the main body in the final version.
>
>
> -- "Why employ a "brute force" approach like RL here? Wouldn't it be better to train the stencils in a supervised manner, e.g., with a more accurate discretization as reference? One could argue that it would be expensive to pre-compute such data, but I think RL scales even worse to higher dimensional problems"
> As already explained in our paper (bullet discussions in section 1.2), the main motivation of using RL is that the problem is naturally a sequential decision making problem. Thus, it can naturally be formulated into a MDP and solved by RL. The main benefit of using RL is that it enforces long term accuracy on the learned policy, making it non-greedy. Since this is a rather important question from the reviewers, we have added more experiments and discussions in the appendix (section A.1) of the revised manuscript. Please let us know if you have further questions.
>
> As for the reviewer's concern that the approach might not be easily translated to the high-dimensional problems, there is actually a simple design for doing so. We can simply use the splitting method for high-dimensional problems, which is essentially applying the trained RL policy alternatively on the one-dimensional problem in each spatial direction.
>
>
> -- "What's also missing in the current version is a more thorough discussion of inference and training performance."
> The training time for the RL policy reported in our paper is roughly one and half a day, using a single custom GTX 1080 GPU. For the inference time, it is true that the computation operations in the trained NN model is much more than that of WENO, but we could parallel and accelerate the computations using GPU, and the real computation time always depends on the implementation.
> We compared four methods: RL-WENO on CPU, RL-WENO on GPU, a well-optimzied WENO (e.g., with good numpy vectorization in python), a poor-implemented WENO (e.g. use lots of loops), and detailed results are reported in table 5 at appendix A.3. The conclusion is: as the grid becomes more dense, all methods except the RL-WENO GPU requires more time to finish the computation. The reason that the time cost of the GPU-version of RL-WENO does not grow up is that on GPU, we can compute all approximations in the next step ($(U^{t+1}_0, U^{t+1}_1, ..., U^{t+1}_J)$) together in parallel, so the increase of grid numbers does not affect the computation time at all. So for coarse grid, well-optimized WENO indeed has clear advantage over RL-WENO (even on GPU), but with a more dense grid, RL-WENO could finish the computation even faster than well-optimzied WENO by leveraging the power of paralleling.

---

### Decision · Program_Chairs · 2019-12-19

**Decision:**

Reject

**Comment:**

This paper proposes using RL to solve PDEs, with application to solving conservation laws. It is quite borderline, with one reviewer weakly recommending acceptance, one finding the paper interesting but the application not sufficiently novel, and one confessing they have not understood the paper.

I concur with R2 this is a difficult subject matter, but the other reviewers seem satisfied with the clarity of the presentation. R3 seems to believe the paper sufficiently proves the concept to warrant publication. I confess I do not understand R1's argument for lack of novelty, despite my pushing for further detail. I see this as a novel application of RL methods, and R1 admits this will be seen as novel for a PDE conference. I am in favour of a certain degree of interdisciplinarity at ICLR, and believe this paper could bring a bit of subject matter diversity to the programme. However, due to the number of high quality submissions in my area, I'm afraid this one must be rejected due to limited space. The authors are encouraged to submit to another ML conference after addressing (or having addressed) some of the action items from the more sympathetic reviewers.